# Fathers involvement in child feeding and its associated factors among fathers having children aged 6 to 24 months in Antsokia Gemza Woreda, Ethiopia: Cross-sectional study

**Solomon Ketema Bogale[1]◉, Niguss Cherie[2]◉, Eyob Ketema Bogale◉[3]◉ \***

1 Antsokiya Gemza Wereda Health Office, North Shoa, North East Ethiopia, 2 Public Health Nutrition Department, Wollo University, Dessie, Ethiopia, 3 Health Promotion and Behavioral Sciences Department, School of Public Health, College of Medicine and Health Sciences, Bahir Dar University, Bahir Dar, Ethiopia

◉ These authors contributed equally to this work.

\* ketema.eyob@gmail.com

**Data Availability Statement:** All relevant data are within the article and its Supporting Information files.

## Abstract

### Background

Father Involvement is exercising positive influences on child feeding. Mothers are usually the primary caregivers for young children. The role of fathers in the proper child feeding of young children has not been a frequent topic of study. Past research has found low rates and little is known about fathers' involvement in child feeding for children 6–23 months in Ethiopia. The aim of this study is to assess fathers' involvement in child feeding children aged 6–24 Months.

### Objectives

To assess fathers' involvement in child feeding and associated factors among fathers having children aged 6 to 24 months.

### Methodology

A community-based cross-sectional study design was conducted from Jan 23/2022 to April 07/2022. A systematic random sampling technique was applied to select study participants. A total of 408 respondents participated in the study. Data was entered into EPI data version 3.1 and then exported to SPSS version 25 for analysis. Bivariable and multivariable logistic regression was used for analysis.

### Result

Father's involvement in child feeding was 43.1%. Factors that were significantly associated with good fathers involvement in child feeding include urban residence(AOR = 3.878, 95% CI = (1.408–10.678), male sex of the youngest child(AOR = 3.681, 95% CI = (1.678–

**Funding:** The authors received no specific funding for this work.

**Competing interests:** The authors have declared that no competing interests exist.

**Abbreviations:** CF, Child Feeding; CHMIS, Community Health Management Information System; CI, Confidence Interval; IYCF, Infant and Young Child Feeding; PAS, Proportional Allocation of the study; SRS, Simple Random Sampling; SPSS, Statistical Package for Social science; WHO, World health organization; IRB, Institution Review Board; ZPGC, Zemen Post Graduate College and; AOR, Adjusted odd ratio.

8.075)), first birth order of the youngest child(AOR = 3.970, 95% CI = (1.212–13.005)), Better(secondary and higher) educational status (AOR = 4.945,95% CI = (1.043–23.454)) and AOR = 5.151, 95% CI = (1.122–23.651)), having ever heard information(AOR = 8.593, 95% CI = (3.044–24.261)), good knowledge (AOR = 3.843,95% CI = (1.318–11.210)), positive attitude (AOR = 8.565, 95% CI = (3.521–20.837)) and good culture (AOR = 10.582,95% CI = (2.818–39.734)).

## Conclusions

Father involvement in child feeding was poor in Antsokia Gemza Woreda. Urban residences, male sex of the youngest child, first birth order of the youngest child, better (secondary and higher) educational status, having ever heard information, good knowledge, positive attitude, and good culture were significantly associated with fathers' involvement in child feeding. Health information dissemination on father involvement in child feeding should be strengthened.

## Introduction

Father involvement is exercising positive influences on child feeding practices and can be involved in financial and resource support; social support and physical support such as shared responsibility for the nutritional well-being and health of the child. Positive nutritional outcomes in children can be brought by father involvement as seen by a study in Ethiopia which found that for children aged between 6–23 months, dietary diversity in the household increased by 13.7%. To promote optimal feeding practices in children, there is a need to increase the direct involvement of fathers [1].

Father Involvement has been shown to significantly improve appropriate infant and young child feeding practices [2]. Father involvement has been associated with positive social, emotional, psychological, developmental, and health outcomes in a child [3]. Proper child feeding is crucial for optimal growth, health, and development of children, especially in the age of 0-2 years. However, this period is often marked by micronutrient deficiencies that result in frequent childhood illnesses [4].

A study in Vietnamese, Peruvian, sub-Saharan Africa, and South Africa showed that children whose fathers were not involved in child feeding were found to be at higher risk of malnutrition, which indicates the need for paternal involvement in child feeding and child health care system in general and nutritional outcome in particular [5].

Globally Over 33% of all infant deaths globally can be indirectly linked to malnutrition. [6]. And an estimated 5.4 million under 5 children died in 2017 and roughly half of these deaths occurred in sub-Saharan Africa [7]. In Ethiopia, 38% of children are stunted, 10% are wasted, and 24% are underweight which informed the occurrence of both acute and chronic undernutrition [8]. Child feeding practice is a significant topic in developing countries where undernutrition is the underlying cause of more than 50% of child deaths [9].

Studies reveal a strong correlation between dads' substantial involvement in infant feeding activities and nutritional diversification [10]. and families with involved fathers had significantly better breastfeeding practices than those with less involved fathers, such as attending breastfeeding sessions with moms, participating in decision-making, and supporting the mother around the house [11, 12]. Another community-based participatory intervention

study revealed that involving both parents equally and actively in child-feeding is a promising strategy for preventing childhood obesity [13]

Most of the studies and interventions on the parents' role in Infant Young Child Feeding (IYCF) have focused on mothers, not fathers [14]. The role of fathers in child health including child feeding has recently begun to appear on the agenda of the developed world. However, in developing countries, despite fathers overruling roles in the incoming generation and decision-making for the family, they have been largely neglected in agendas for improvements to child health [15–17].

In Ethiopia, there are gender-specific roles. Fathers are responsible for raising money and ensuring there is food from the garden for the family, but looking after the children and performing domestic activities are mothers' responsibilities. Mothers are usually considered to be subordinate to their husbands [18].

Interventions have been implemented through mothers as the primary caregivers to combat child malnutrition, but the father's role is usually considered insignificant even though a child's full development probably depends on the complex care that they receive from both the mother and father [19].

Determining the extent of Father Involvement in child feeding is important because early childhood represents a key developmental window to shape healthy behaviors into late childhood and adulthood [20]. A study in Ethiopia shows that fathers who had good participation in child feeding activities have a positive influence on better dietary diversity [21].

Alive &Thrive's activities in Ethiopia are designed to engage fathers in child feeding by identifying the strategies that seem to make these programs work to ensure that fathers can play an active role in improving child feeding practice by applying 6 strategies such as grabbing their attention with emotion, ease the way by busting stereotypes, find fathers where they already are, "provide crystal-clear direction" for actions fathers can take, give fathers practice and show fathers a benefit that they care about [22].

To date studies most often address fathers' socioeconomic status in relation to children's lives, but they are superficial and insufficient evidence for understanding fathers' involvement in child feeding practice. [23, 24]. To the best of our knowledge, there is minimal scientific literature on fathers' involvement in child feeding. However, their role in their family other than income generation, farming, and dealing with family contacts outside the home is very limited.

The finding of this study will be helpful for policymakers (whether governmental or Non-Governmental Organizations working on child feeding practices) to design evidence-based alternative strategies. It is also used by researchers as a baseline for future related studies. It is also key for health professionals for providing evidence-based counseling on the study area. The purpose of this study will be to assess fathers' involvement in child feeding practice and associated factors among fathers having children aged 6 to 24 months to address the aforementioned gap in the area.

## Materials and methods

### Study design and settings

A community-based cross-sectional study design was conducted from Jan 23 to April 07, 2022, among fathers who had children aged between 6 -and 24 months and residing in Antsokiya Gemza Woreda (Fig 1). it is found in the North Shoa zone, Amhara region around 400 km far away from Addis Ababa, 505 km from the city of Amhara regional state, which is Bahir Dar city, and 270 km from zonal capital Debre Berhan. The estimated total population of the woreda is estimated to be 66673 in 2021: among those 20% live in urban and 80% of them in rural

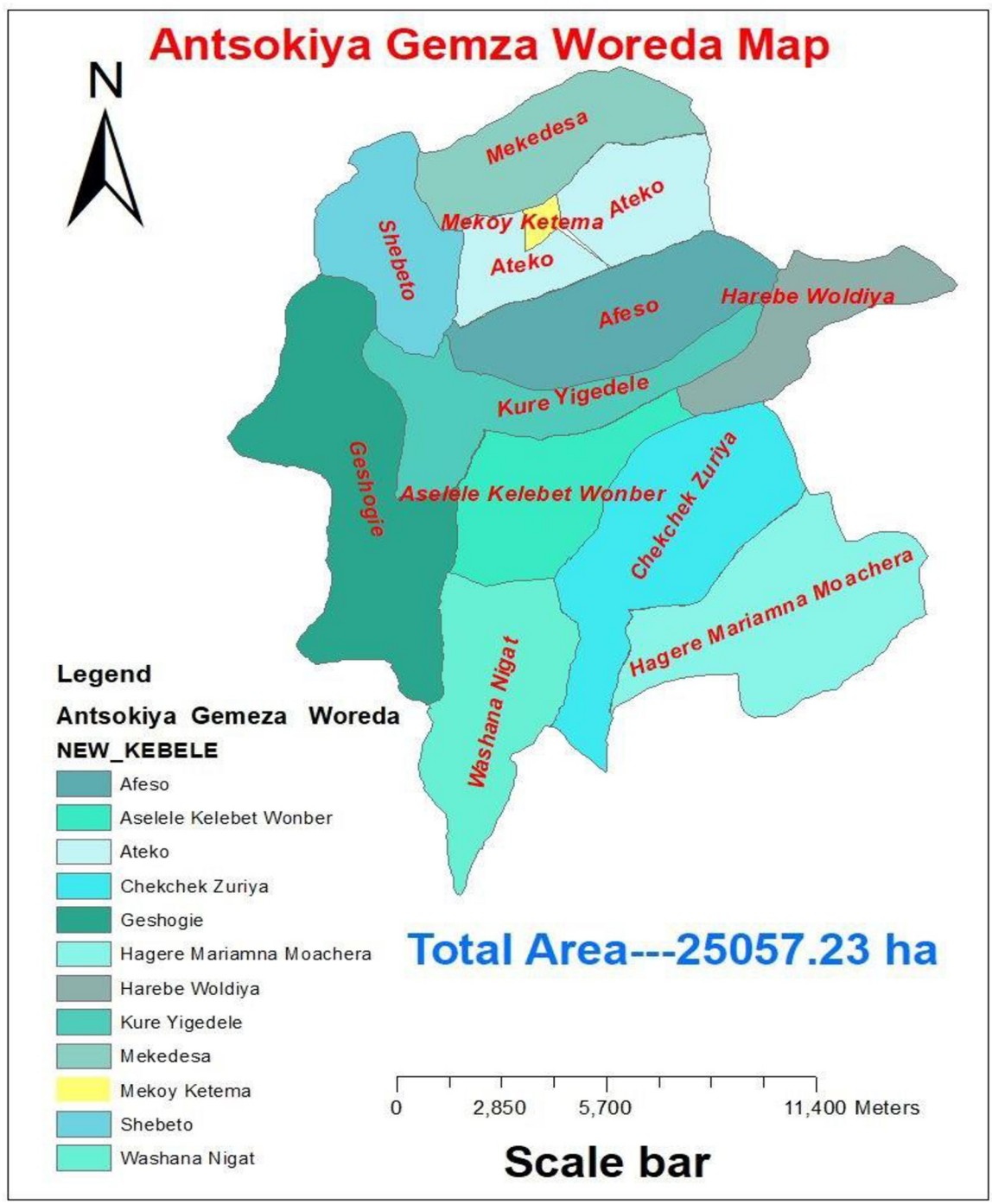

**Fig 1. Map of Antsokia Gemza Woreda.**

areas. The majority of the inhabitants are orthodox which is 85%, 10% are Muslim and the rest 5% are protestant by religion. The woreda has 11% cold, 44% moderate, and 54% hot weather conditions. According to the 2021 report of the Antsokia health sector office, there were 5 health centers and 12 health posts. There are also around 2927 fathers who have children from 6 to 24 months living in the woreda [25].

## Participants and sampling

Fathers who had children aged 6 to 24 months and household members, who resided in Antso-kiya Gemza Woreda for at least six months during data collection were included in the study whereas fathers who were critically ill and unable to communicate during the data collection period were excluded from the study.

The sample size was calculated by considering the two specific objectives; using the formula for estimation of single population proportion and using Epi info version 7 to identify the fac-tors. Based on the single population proportion assumptions were: A 95% confidence level (Z), 5% margin of error (E), and the proportion of fathers involved in child feeding was 59.1% (from the study in Dakshina Kannada, a coastal district in Karnataka State) in the previous study [26]. The final sample size was 408 after adding 10% non-respondent. We took the larg-est estimated sample.

Systematic random sampling was applied to select study participants. All 12 kebeles were incorporated into the study. The data were collected within a one-month duration. Propor-tional allocation of the study (PAS) participants was considered among all kebeles. A list of households with the study subjects was identified using the community health management information system (CHMIS) folder in the health post for rural kebeles and through census for urban kebele. The sampling interval (kth unit) was obtained by dividing the entire fathers (the total number of fathers who had children from 6 to 24 months) (2927) by the desired sample size (total number of sample size (408) and it was approximately 7. The first father was ran-domly chosen for the survey by the lottery method from the first seven fathers, and then every seventh was recruited for the study to identify a total of 408 samples. The name and address of fathers having children aged 6–24 months were specified and their location was identified in collaboration with the kebele's health extension workers and health development army leaders, identified fathers were then interviewed in their homes.

## Data collection

Literature is reviewed to develop a questionnaire [27]. The data were collected using pretested, structured, interviewer-administered questionnaire. Interviews were conducted at the father's home in the Amharic language. Ethical approval for this study was obtained from the Institu-tion Review Board (IRB) of Zemen Post Graduate College. A permission letter from the Antso-kia Gemza Woreda Health office and selected kebeles administrations were informed about the study. Data were collected after explaining the study, the benefit of the study, the research purpose orally, and getting verbal informed consent from each participant. Data were collected by four clinical nurses. Data collectors were trained for one day prior to the data collection. Two Health officers supervised the data collection. Each day questionnaires were checked for completeness and consistency by the principal investigator and supervisors.

## Measurements

Fathers' involvement in child feeding practice was assessed using 22 items with five major domains such as shared decision making in child feeding practice (6 items), providing physical support to the mother (5 items), providing psychosocial support (3 items), providing financial and resource support (5 items) and workload sharing (3 items). From these 22 questions directed to the fathers, variable scores of 12 and above were considered as good fathers' involvement while variable scores of below 12 were considered as poor fathers' involvement in child feeding practices [28].

Knowledge of fathers towards fathers' involvement in child feeding was also assessed using 9 items that were directed to the fathers. The questions asked had the following components

asked in such a way that do you know about; baby start complementary feeding at 6 months, recommended duration of breastfeeds 24 months or more, have a role in providing advice on diet, have a role in farming/gardening nutritious food, have a role in financial support to buy nutritious food, have a role in cooking a meal for a child, have a role in accompaniment to medical appointments/growth monitoring, have a role in social and emotional support and have a role to help child feeding mothers with household chores. Fathers who scored five and above among the listed nine items were considered as having good knowledge of fathers' involvement in child feeding practices while those who scored below five were considered as having poor knowledge of fathers' involvement in child feeding practices [29].

The responses of fathers about the attitude towards father involvement in child feeding were in the form of a 5-point Likert scale, wherein a score of +2 meant they strongly agreed with the statement, +1 meant agreed, 0 neutral feeling to the statement, -1 meant disagreed to the statement and -2 was the score for strongly disagree. The total score was calculated and transformed into the mean 'percent score' by dividing the score by the possible maximum score and multiplying it by 100. Scores for attitude were categorized as negative [0-< = 60%] and positive [>60%] [29].

## Data quality assurance

A pre-test was done on 5% (20) of the sample size before the actual data collection. Amendments were made on questions after pre-testing. The questionnaire was prepared first in English and then translated to the local language of Amharic and back-translated to English by a third person who was native to Amharic and had experience in translation. The reliability tests were checked using Cronbach's alpha of 0.7 as the cut-off point to assess the internal consistency of the research instrument. The overall Cronbach's alpha coefficient value for the data collecting instrument was found to be 0.827 indicating the acceptability of the scale for further analysis. Hosmer and Lemeshow's test value was checked for its model fitness and its result is 0.833 indicating the model was fit.

The training was given to data collectors about the data collection tool and how to collect data before data collection, to have a common understanding. Data were cleaned and checked for outliers before analysis. Negatively worded statements of the items were reverse coded before analysis.

Literature is reviewed to develop a questionnaire [27] and it was modified in our context. Furthermore, we have done a pretest since pre-tests also provide the most direct evidence for the validity of the questionnaire. Moreover, to determine the content validity of the questionnaire, the collected views, as well as comments from experts in the fields of nutrition, biostatics, and epidemiology, were taken.

## Data management and analysis

The data were collected, checked for completeness and consistency, and then entered into epi data version 4.7. The data was coded and cleaned; finally, it was exported to Statistical Package for Social Science (SPSS) version 25 software. The data were checked for outliers and missing values and were analyzed using SPSS version 25 software. Descriptive statistics were used to describe the percentage and number of distribution of respondents by each variable. Descriptive summary measures such as mean and median were computed and the results were presented using texts, figures, and tables.

Before logistic regression analysis, the assumption was checked and the data qualified for logistic regression. Bivariable and multivariable logistic regression analysis was used to identify predictors of behavioral responses. Independent variables with a P-value < of 0.25 in the

bivariate analysis were entered into the multivariable logistic regression in order to control the possible effect of confounders using the backward likelihood regression variable selection method.

In the final model, a p-value < 0.05 was considered statistically significant. Significant independent variables were declared at a 95% Confidence Interval/CI/ and a P-value of less than 0.05. The odds ratio is used to observe the strength of the association between a dependent variable and each significant independent variable. In the multivariable model, the independent variables with a P-value < of 0.05 and AOR with a 95% confidence interval were used to set the statistically significant level and to identify predictors of fathers' involvement in child feeding.

## Results

### Socio-demographic characteristics

A total of 408 fathers who had children between 6 to 23 months were enrolled in the study with a response rate of 100%. This 100% response rate was achieved by multiple contacts; a person was deemed non-responsive if absent after three consecutive visits. The mean age of the study participants was 36.8 with a standard deviation (SD) of 8.892 years and the majority of them, 334 (81.9%) lived in rural areas (Table 1).

### Source of information of respondents about father involvement in child feeding

The majority 309(75.7%) of the respondents heard about father involvement in CF, from those about 177(43.4%) of them heard from health extension workers and health professionals, 94 (23%) from television, 60(14.7%) from radio, and 38(9.3%) from relative/friend/family.

### Knowledge of fathers about father involvement in child feeding

The majority, 323 (79.2%) of participants realized that babies should start eating foods in addition to breastmilk at 6 months of age. About 294(72.1%) of respondents identify fathers' involvement in child feeding by providing diet advice followed by attending medical appointments/growth monitoring 271(66.4%). Overall, about 209 (51.2%) of fathers had poor knowledge and 199(48.8%) had good knowledge about father involvement in child feeding (Table 2).

### The attitude of fathers about father involvement in child feeding

Nearly half of the respondents 202(49.5%) agreed and 43(10.5%) strongly agreed that confident in preparing food for their child. About 185(45.3%) of respondents agreed and 89(21.8%) strongly agreed that feel confident that supporting their wife with child feeding and care. Overall, 228 (55.9%) fathers had a negative attitude, or 180(44.1%) had a positive attitude toward father involvement in child feeding (Table 3).

### Culture of fathers about father involvement in child feeding

About 226(55.4%) of respondents replied that the community didn't discourage fathers to take an active role in child feeding, and also wives want fathers' involvement in child feeding 285 (69.6%), the community didn't laugh, sanctioned in father involvement in child feeding 255 (62.5%), there was no traditionally gender-specific role in the community 212(52%), and community didn't think, child feeding is the only responsibility of mother 214(52.5%). Overall, 266 (65.2%) of respondents had good culture toward father involvement in child feeding (Table 4).

**Table 1. Socio-demographic characteristics of father involvement in child feeding among fathers who had children aged from 6–24 months in Antsokia Gemza Woreda,2022 (n = 408).**

| Sociodemography Characterstics | Frequency | Percent |
|---|---|---|
| Age of fathers(In years) | | |
| Mean age | 36.8 | |
| 15–20 | 20 | 4.9 |
| 21–30 | 94 | 23.0 |
| 31–40 | 166 | 40.7 |
| 41–50 | 104 | 25.5 |
| 50+ | 24 | 5.9 |
| Residence | | |
| Urban | 74 | 18.1 |
| Ruralg | 334 | 81.9 |
| Religion | | |
| Orthodox | 236 | 57.8 |
| Muslim | 127 | 31.1 |
| Protestant | 45 | 11.0 |
| Number of children | | |
| < = 4 | 329 | 80.6 |
| >4 | 79 | 19.4 |
| Sex of the youngest child | | |
| Male | 233 | 57.1 |
| Female | 175 | 42.9 |
| Age of the youngest child in month | | |
| 6–12 | 150 | 36.8 |
| 12–18 | 91 | 22.3 |
| 18–24 | 167 | 40.9 |
| Birth order of the youngest child | | |
| First child | 71 | 17.4 |
| Not first child | 337 | 82.6 |
| Marital status of Father | | |
| Married | 345 | 84.6 |
| Divorced | 45 | 11.0 |
| Widowed | 18 | 4.4 |
| Educational status of Father | | |
| Not formal education | 169 | 41.4 |
| Primary education | 170 | 41.7 |
| Secondary education | 44 | 10.8 |
| Higher education(12+) | 25 | 6.1 |
| Occupational status of Father | | |
| Farmer | 314 | 77.0 |
| Merchant | 33 | 8.1 |
| Government | 48 | 11.8 |
| Daily laborer | 13 | 3.2 |
| Household monthly income In Ethiopian birr | | |
| LESSTHAN 500 | 31 | 7.6 |
| 500–1000 | 45 | 11.0 |
| 1000–1500 | 38 | 9.3 |

(*Continued*)

**Table 1.** (Continued)

| Sociodemography | | |
| --- | --- | --- |
| Characterstics | Frequency | Percent |
| 1500–2000 | 29 | 7.1 |
| 2000–2500 | 38 | 9.3 |
| 2500–3000 | 84 | 20.6 |
| 3000–4000 | 97 | 23.8 |
| 4000+ | 46 | 11.3 |

Note: 52.811 Ethiopian birr is equivalent to 1 USD dollars, minimum wage for a family to classify as poor is less than 500 birrs.

## The magnitude of father involvement in child feeding

The magnitude of the father's involvement in child feeding was found to be 43.1% good with 95% CI (38.7, 48.3) and 56.9% poor with 95% CI (51.7, 61.3) (Table 5).

**Table 2. Knowledge of fathers about father involvement in child feeding in Antsokia Gemza Woreda, 2022.**

| Father Knowledge | Frequency | Percent |
| --- | --- | --- |
| An infant should start complementary food at 6 months | | |
| Yes | 323 | 79.2 |
| No | 85 | 20.8 |
| A woman should breastfeed her child for 24 months and more | | |
| Yes | 282 | 69.1 |
| No | 126 | 30.9 |
| Do you know fathers have role in providing advice for mothers on child diet | | |
| Yes | 294 | 72.1 |
| No | 114 | 27.9 |
| Do you know fathers have role in farming/gardening nutritious food | | |
| Yes | 243 | 59.6 |
| No | 165 | 40.4 |
| Do you know fathers have role in financial support to buy nutritious food | | |
| Yes | 253 | 62.0 |
| No | 155 | 38.0 |
| Do you know fathers have role in cooking a meal for a child | | |
| Yes | 242 | 59.3 |
| No | 166 | 40.7 |
| Do you know fathers have role in accompaniment to medical appointments/growth monitoring | | |
| Yes | 271 | 66.4 |
| No | 137 | 33.6 |
| Do you know fathers have role in in social and emotional support | | |
| Yes | 256 | 62.7 |
| No | 152 | 37.3 |
| Do you know fathers have role to help child feeding mothers with house hold chores | | |
| Yes | 265 | 65.0 |
| No | 143 | 35.0 |
| Knowledge of Father | | |
| Good knowledge | 199 | 48.8 |
| Poor knowledge | 209 | 51.2 |

**Table 3. Attitude of fathers about father involvement in child feeding in Antsokia Gemza Woreda, 2022.**

| | Strongly disagree | | Disagree | | Neutral | | Agree | | Strongly agree | |
|---|---|---|---|---|---|---|---|---|---|---|
| | Frequency | % | Frequency | % | Frequency | % | Frequency | % | Frequency | % |
| I am confident in preparing food for my child. | 24 | 5.9 | 108 | 26.5 | 31 | 7.6 | 202 | 49.5 | 43 | 10.5 |
| I feel difficult in giving the right kind of food for my child. | 69 | 16.9 | 173 | 42.4 | 17 | 4.2 | 131 | 32.1 | 18 | 4.4 |
| I feel less important for a father to spend much time with my children. | 102 | 25.0 | 174 | 42.6 | 8 | 2.0 | 107 | 26.2 | 17 | 4.2 |
| I feel difficult that support my wife for child feeding and care. | 78 | 19.1 | 156 | 38.2 | 26 | 6.4 | 121 | 29.7 | 27 | 6.6 |
| I feel confident that support my wife for child feeding and care. | 4 | 1.0 | 97 | 23.8 | 33 | 8.1 | 185 | 45.3 | 89 | 21.8 |
| I feel right that Mother should be as heavily involved in the child feeding than fathers. | 9 | 2.2 | 30 | 7.4 | 7 | 1.7 | 234 | 57.4 | 128 | 31.4 |
| I feel happy, if my wife asked me to help her by feeding the child. | 11 | 2.7 | 87 | 21.3 | 24 | 5.9 | 185 | 45.3 | 101 | 24.8 |
| I feel correct that child feeding practice is a shared responsibility of father and Mother | 10 | 2.5 | 55 | 13.5 | 12 | 2.9 | 168 | 41.2 | 163 | 40.0 |
| Attitude of Father | | | Frequency | | | | Percent | | | |
| Positive attitude | | | 180 | | | | 44.1 | | | |
| Negative attitude | | | 228 | | | | 55.9 | | | |

## Factors associated with father involvement in child feeding

In the bivariate analysis, age of the father, residence, number of children, sex of the youngest child, age of the youngest child, birth order of the youngest child, marital status, educational status, occupational status, monthly income, heard about father involvement, knowledge, attitude and culture were significantly associated with father involvement in child feeding. And hence, entered into the multivariable regression model.

After adjusting the effect of other variables (confounders), using multivariable logistic regression, residence, sex of the youngest child, birth order of the youngest child, educational status, heard about father involvement, knowledge, attitude, culture was significantly associated with fathers' involvement in child feeding.

**Table 4. Culture of fathers about father involvement in child feeding in Antsokia Gemza Woreda, 2022.**

| Culture items | Frequency | Percent |
|---|---|---|
| Community discourage father to take an active role in CF | | |
| Yes | 182 | 44.6 |
| No | 226 | 55.4 |
| Your wife did not want father involvement in CF | | |
| Yes | 123 | 30.1 |
| No | 285 | 69.9 |
| community laughed, sanctioned in your involvement in CHF | | |
| Yes | 153 | 37.5 |
| No | 255 | 62.5 |
| traditionally gender specific role in Your community | | |
| Yes | 195 | 47.8 |
| No | 213 | 52.2 |
| community think, CF is the only responsibility of mother | | |
| Yes | 194 | 47.5 |
| No | 214 | 52.5 |
| Culture of Father | | |
| Good culture | 266 | 65.2 |
| Bad culture | 142 | 34.8 |

**Table 5. Father involvement in child feeding in Antsokia Gemza Woreda, 2022.**

| Father involvement activities in CF | Frequency | Percent |
|---|---|---|
| discussing with your wife in CF before you become a decision | | |
| Yes | 265 | 65.0 |
| No | 143 | 35.0 |
| father have equal decision making as mothers at home | | |
| Yes | 239 | 58.6 |
| No | 169 | 41.4 |
| final decision on childfeeding | | |
| Yes | 218 | 53.4 |
| No | 190 | 46.6 |
| final decision on time to start complementary feeding | | |
| Yes | 224 | 54.9 |
| No | 184 | 45.1 |
| final decision on what food for start of complementary feeding | | |
| Yes | 229 | 56.1 |
| No | 179 | 43.9 |
| final decision on order of serving food during meal times | | |
| Yes | 188 | 46.1 |
| No | 220 | 53.9 |
| participate in child feeding during meal times | | |
| Yes | 239 | 58.6 |
| No | 169 | 41.4 |
| assist mother with household chores | | |
| Yes | 227 | 55.6 |
| No | 181 | 44.4 |
| assist mother with farming activities to get nutritious food for your child | | |
| Yes | 269 | 65.9 |
| No | 139 | 34.1 |
| accompanying mother for child health clinics | | |
| Yes | 249 | 61.0 |
| No | 159 | 39.0 |
| allowing other family members/relatives to support the mother after delivery | | |
| Yes | 280 | 68.6 |
| No | 128 | 31.4 |
| the community encourage you to take an active role in CF | | |
| Yes | 209 | 51.2 |
| No | 199 | 48.8 |
| encourage your children to take food while the mothers were in CF | | |
| Yes | 275 | 67.4 |
| No | 133 | 32.6 |
| motivate your spouse to get involved in the CF | | |
| Yes | 226 | 55.4 |
| No | 182 | 44.6 |
| buying food for the child | | |
| Yes | 259 | 63.5 |
| No | 149 | 36.5 |
| buy clothing, child care items like diapers and d/t child food for your child | | |
| Yes | 239 | 58.6 |

(*Continued*)

**Table 5.** (Continued)

| Father involvement activities in CF | Frequency | Percent |
|---|---|---|
| No | 169 | 41.4 |
| buying food for the lactating mother | | |
| Yes | 276 | 67.6 |
| No | 132 | 32.4 |
| transporting to the child to health clinics | | |
| Yes | 260 | 63.7 |
| No | 148 | 36.3 |
| gave money to the mothers to purchase the necessary food for the children | | |
| Yes | 286 | 70.1 |
| No | 122 | 29.9 |
| you usually feed the child at home | | |
| Yes | 104 | 25.5 |
| No | 304 | 74.5 |
| cook a meal for a child food at home, when a mother is in breast feeding | | |
| Yes | 194 | 47.5 |
| No | 214 | 52.5 |
| looked after Your child when the mothers were not around | | |
| Yes | 244 | 59.8 |
| No | 164 | 40.2 |
| Father involvement | | |
| Good | 176 | 43.1 |
| Poor | 232 | 56.9 |

Fathers who lived in urban areas were 3.878 times more likely to have good involvement in child feeding than those fathers who lived in rural areas (AOR = 3.878, 95% CI = (1.408–10.678)). Those fathers who had a male youngest child were 3.681 times more likely to have good involvement in child feeding than those who had a female youngest child (AOR = 3.681, 95% CI = (1.678–8.075)). Those fathers whose childbirth order is first were 3.970 times more likely to have good involvement in child feeding than those whose child was not first child (AOR = 3.970, 95% CI = (1.212–13.005)).

Fathers who attended secondary education were 4.945 times more likely to have good involvement in child feeding than those fathers who were not in normal education (AOR = 4.945, 95% CI = (1.043–23.454)). Fathers who attended higher education were 5.151 times more likely to have good involvement in child feeding than those fathers who were not in normal education (AOR = 5.151, 95% CI = (1.122–23.651)).

Those fathers who had ever heard information were 8.593 times more likely to have good involvement in child feeding than had not ever heard information about father involvement in child feeding (AOR = 8.593, 95% CI = (3.044–24.261)).

Those fathers who had good knowledge were 3.843 times more likely to have good involvement in child feeding than those who had poor knowledge (AOR = 3.843, 95% CI = (1.318–11.210)). Those fathers who had positive attitudes were 8.565 times more likely to have good involvement in child feeding than those who had negative attitudes (AOR = 8.565, 95% CI = (3.521–20.837)). Those fathers who had good culture were 10.582 times more likely to have good involvement in child feeding than those who had bad culture (AOR = 10.582, 95% CI = (2.818–39.734)) (Table 6).

**Table 6. Multivariable logistic regression model for fathers' involvement in child feeding practice in Antsokia Gemza Wereda, North East Ethiopia, 2022.**

| Variables | Father involvement in child feeding | | COR (95%) | AOR (95%) |
| --- | --- | --- | --- | --- |
| Characterstics | Poor | Good | | |
| age of fathers in years | | | | |
| 15–20 | 5 | 15 | 3.545(.974–12.905) | .122(.012–1.250) |
| 21–30 | 52 | 45 | 1.023(.417–2.507) | .150(.022–1.031) |
| 31–40 | 93 | 73 | .928(.393–2.191) | .176(.028–1.118) |
| 41–50 | 69 | 32 | .548(.222–1.356) | .178(.028–1.141) |
| 50+ | 13 | 11 | 1 | 1 |
| Residence | | | | |
| Urban | 13 | 61 | 8.936(4.712–16.944) | 3.878(1.408–10.678)** |
| Rural | 219 | 115 | 1 | 1 |
| number of children | | | | |
| < = 4 | 178 | 151 | 1.832(1.088–3.086) | .885(.301–2.602) |
| >4 | 54 | 25 | 1 | 1 |
| Sex of the youngest child | | | | |
| Male | 114 | 119 | 2.161(1.438–3.248) | 3.681(1.678–8.075)** |
| Female | 118 | 57 | 1 | 1 |
| Age of the youngest child in month | | | | |
| 6–12 | 74 | 76 | 1.423(.913–2.218) | 1.013(.451–2.274) |
| 12–18 | 61 | 30 | .681(.399–1.163) | .713(.224–2.277) |
| 18–24 | 97 | 70 | 1 | 1 |
| Birth order of the youngest child | | | | |
| First child | 25 | 46 | 2.930(1.717–4.998) | 3.97(1.212–13.005) ** |
| Not first child | 207 | 130 | 1 | 1 |
| Marital status of Father | | | | |
| Married | 188 | 157 | 4.176(1.187–14.69) | .326(.041–2.576) |
| Divorced | 29 | 16 | 2.759(.693–10.984) | .145(.013–1.559) |
| Widowed | 15 | 3 | 1 | 1 |
| Educational status of Father | | | | |
| Not formal education | 132 | 37 | 1 | 1 |
| Primary education | 81 | 89 | 3.920(2.443–6.289) | 1.368(.581–3.222) |
| Secondary education | 12 | 32 | 9.514(4.462–20.282) | 4.945(1.043–3.454)** |
| Higher education(12+) | 7 | 18 | 9.174(3.562–23.628) | 5.151(1.122–3.651)** |
| Occupational status of Father | | | | |
| Farmer | 188 | 126 | 3.686(.803–16.912) | .720(.067–7.772) |
| Merchant | 24 | 9 | 2.062(.380–11.180) | .358(.022–5.799) |
| Government | 9 | 39 | 23.83(4.478–26.85) | 1.238(.097–15.801) |
| Daily laborer | 11 | 2 | 1 | 1 |
| House hold monthly income In Ethiopian birr | | | | |
| LESSTHAN 500 | 28 | 3 | 1 | 1 |
| 500–1000 | 31 | 14 | 4.215(1.095–16.22)* | 1.752(.200–15.38) |
| 1000–1500 | 27 | 11 | 3.802(.955–15.141) | .784(.078–7.921) |
| 1500–2000 | 21 | 8 | 3.556(.840–15.044) | 1.004(.089–11.38) |
| 2000–2500 | 20 | 18 | 8.400(2.177–32.41) | .976(.100–9.503) |
| 2500–3000 | 39 | 45 | 10.77(3.038–38.18) | 1.052(.149–7.445) |
| 3000–4000 | 48 | 49 | 9.528(2.715–33.44) | 1.624(.236–11.17) |
| 4000+ | 18 | 28 | 14.519(3.84–54.88) | 1.892(.233–15.36) |
| Heared about father involvement | | | | |

*(Continued)*

**Table 6.** (Continued)

| Variables | Father involvement in child feeding | | COR (95%) | AOR (95%) |
|---|---|---|---|---|
| **Characterstics** | **Poor** | **Good** | | |
| Yes | 145 | 164 | 8.200(4.308–15.607) | 8.593(3.044–24.26) ** |
| No | 87 | 12 | 1 | 1 |
| Knowledge of Father | | | | |
| Good | 53 | 146 | 16.436(9.986–27.053) | 3.843(1.318–11.21) ** |
| Poor | 179 | 30 | 1 | 1 |
| Attitude of Father | | | | |
| Positive | 33 | 147 | 30.567(17.769–2.585) | 8.565(3.521–20.84) ** |
| Negative | 199 | 29 | 1 | 1 |
| Culture of Father | | | | |
| Good | 95 | 171 | 49.32(19.519–24.621) | 10.582(2.82–39.73) ** |
| Bad | 137 | 5 | 1 | 1 |

Note:

* significant variables at P value <0.25 in the bivariate analysis

** statistically significant variables at P value <0.05 in the multivariable analysis, COR: crude odds ratio; AOR, adjusted odds ratio; 1.00 _ reference category, 52.811 Ethiopian birr is equivalent to 1 USD dollars, minimum wage for a family to classify as poor is less than 500 birrs.

## Discussion

The result of this study revealed that 43.1% of fathers had good involvement in child feeding which is lower than compared the studies conducted in a rural Southwestern District of Uganda, in which 65.5% of fathers had good involvement [30] and in Dakshina Kannada District in South India, in which 59.1% fathers had good involvement in child feeding [26]. This discrepancy might be due to the fact that mothers have primary responsibility for child feeding and the fathers' role often comes after, and children spend most of the time with their mothers. Additionally, fathers frequently work away from their homes making them less accessible to participate in child feeding activities in the study area.

Activities by Alive &Thrive in Ethiopia are intended to get fathers involved in child feeding by identifying and using 6 strategies such as getting their attention with emotion, making the process easier by shattering stereotypes, finding fathers where they already are, "providing crystal-clear direction" for actions fathers can take, give fathers practice and show fathers a benefit that they care about [22].

In this study, fathers who lived in the urban area showed a significant association with good father involvement in child feeding. This finding is contrary to the finding from a district in coastal South India, in which fathers residing in urban locations had higher levels of poor involvement in IYCF than their counterparts [26]. This discrepancy might be due to fathers who lived in the urban area having a good opportunity for media to get full information about father involvement in child feeding than lived in rural areas.

In this study, fathers who had male index children were more likely to get involved in child feeding. This finding is contrary to the finding from South India, in which fathers with male children showed a significantly higher level of poor involvement than those with female children [26]. This discrepancy might be due to fathers being happier when their children's sex would be male than female in the study area.

In this study, being first-child birth order was significantly associated with good father involvement in child feeding. This finding is in agreement with the study conducted in South India, in which fathers who had a first-time child had better involvement than fathers with

more than one child [26]. This might be due to fathers having got children for the first time since they would be happy and try to fulfill all that the child needs.

The educational status of fathers was another independent predictor of fathers' involvement in child feeding practices in that fathers who attended secondary education and higher were more likely to be involved in child feeding practices. This finding is in line with the studies conducted in Nepal and Equatorial Guinea where fathers who had higher levels of education showed greater involvement in encouraging their spouses to breastfeed their children [31, 32]. This might be explained by the fact that fathers who had better education could have better information and a better understanding of the importance of father involvement in child feeding.

The result of our study revealed that fathers who had ever heard information about child feeding were significantly associated with good father involvement in child feeding. This finding is supported by Ethiopian National Strategy on Infant and Young Child Feeding shows that Health professionals, and educational and media authorities provide information about appropriate child feeding practices for fathers, mothers, and other caregivers [33].

In this study, fathers who had good knowledge were significantly associated with good father involvement in child feeding. This finding is in line with the studies conducted in Brazil, Australia, Malaysia, and the United States of America were fathers who had better knowledge of breastfeeding and complementary feeding increased participation in child feeding [34–37]. This might be the fact that fathers had good knowledge affects fathers' attitude, which leads to a change of behavior in good fathers' involvement in child feeding [38]. In addition, the behavior changes communication (BCC) strategies program in Ethiopia that targets fathers' engagement in child feeding in both father and mother has a greater impact on the father's knowledge [39].

The result of our study revealed that fathers who had a positive attitude about father involvement in child feeding were significantly associated with good father involvement in child feeding. This finding is in line with the studies conducted in England and Northern Jordan where fathers who had positive attitudes about child feeding increased fathers' involvement in child feeding [40, 41]. This might be due to a result of a change in behavior in good fathers' involvement in child feeding brought on by a positive attitude [38]. This finding is also supported by the Alive & Thrive strategy in Ethiopia showing that successfully used positive emotions/feelings/attitudes to draw fathers into child feeding [22].

In this study, fathers who had good culture were significantly associated with good father involvement in child feeding. This finding is in line with the studies conducted in Taiwan and Malawi where fathers who had good cultural practices can positively influence parental involvement in child feeding [37,42] These similarities might be due to the community encouraging fathers to participate in child feeding in the study area. This finding is supported by the Alive & Thrive strategy/initiatives in Ethiopia to involve fathers in child feeding shows that the strategy eases the way by busting stereotypes in which gender roles are deeply embedded in culture [22].

## Limitations and strengths of the study

Since the study is cross-sectional it may not demonstrate direct cause and effect between dependent and independent variables. Another limitation is possible social desirability bias despite caution being used in order to reduce it. However, our study has some strengths. As the Ethiopian national nutrition program is targeting children and mothers to overcome malnutrition, this study tried to address father involvement in child feeding practices which if promoted could improve the overall health of children.

## Conclusions

The finding of this study showed that good father involvement in child feeding was low (43.1%), since the world's Father's Day 2019 report, which aims to facilitate an enabling environment where men take on 50% of involvement. Better (secondary and higher) educational status, having ever heard (information), urban residence, male sex of the youngest child, first birth order of the youngest child, good knowledge, positive attitude, and good culture were predictors of good fathers' involvement in child feeding. The purpose of this study was to assess fathers' involvement in child feeding practice and associated factors among fathers having children aged 6 to 24 months Hence, it is important to promote fathers' involvement in child feeding practices through promoting their education, availing information sources, improving their knowledge, attitude and culture of fathers about child feeding practices. Future researchers should better conduct studies by using a mixed methodology and including females as potential participants to ensure fathers' involvement from women's perspectives.

## Supporting information

**S1 File.**
(DOCX)

## Acknowledgments

We would like to thank the Antsokia Gemza Health office and each kebele administration for giving valuable information and permission. We want to give our special thanks to the study participants for their willingness to participate, data collectors, and supervisors. Finally, it is our pleasure to give our deepest thanks to our family for their contributions and patience throughout this study.

## Author Contributions

**Conceptualization:** Solomon Ketema Bogale, Niguss Cherie, Eyob Ketema Bogale.

**Data curation:** Solomon Ketema Bogale, Niguss Cherie, Eyob Ketema Bogale.

**Formal analysis:** Solomon Ketema Bogale.

**Investigation:** Solomon Ketema Bogale.

**Methodology:** Solomon Ketema Bogale, Niguss Cherie, Eyob Ketema Bogale.

**Project administration:** Solomon Ketema Bogale.

**Resources:** Solomon Ketema Bogale.

**Software:** Solomon Ketema Bogale, Niguss Cherie, Eyob Ketema Bogale.

**Supervision:** Niguss Cherie, Eyob Ketema Bogale.

**Validation:** Solomon Ketema Bogale, Niguss Cherie, Eyob Ketema Bogale.

**Visualization:** Niguss Cherie, Eyob Ketema Bogale.

**Writing – original draft:** Solomon Ketema Bogale, Niguss Cherie, Eyob Ketema Bogale.

**Writing – review & editing:** Solomon Ketema Bogale, Niguss Cherie, Eyob Ketema Bogale.

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
