## [Decision Letter · Decision Letter 0]

6 Jul 2022

PONE-D-22-14665Fathers Involvement in Child Feeding and Its Associated Factors Among Fathers Having Children Aged 6 to 24 Months in Antsokia Gemza Woreda, Ethiopia: Cross-Sectional StudyPLOS ONE

Dear Dr. Bogale,

Thank you for submitting your manuscript to PLOS ONE. After careful consideration, we feel that it has merit but does not fully meet PLOS ONE’s publication criteria as it currently stands. Therefore, we invite you to submit a revised version of the manuscript that addresses the points raised during the review process.

We look forward to receiving your revised manuscript.

Kind regards,

Khatijah Lim Abdullah, DClinP, MSc., BSc

Academic Editor

PLOS ONE

Journal Requirements:

 [No funders]. 

5. Please ensure that you refer to Figure 3 in your text as, if accepted, production will need this reference to link the reader to the figure.

6. Please include your tables as part of your main manuscript and remove the individual files. Please note that supplementary tables (should remain/ be uploaded) as separate "supporting information" files.

Additional Editor Comments:

Please ensure the manuscript has undergo proof reading before resubmission 

 Review Comments to the Author

Reviewer #1: 1) Background - is there any studies as such has been identified in the world health report? The write up needs to be reword and revised such as too many dash in page 4. It needs to be written clearly

2) methods section, how did the author reach the community? Please explain. What is the response rate? What is kebeles in page 5.

- With the questionnaire that is used has it been validated especially the KAP or etc. Please cite it. If pre test done, is there any variables been removed? For data analysis, author highlighted double entry. Did you do that and what percentage is the discrepancy?

- Ethical approval?

3) the discussion is too long. What is the limitations and strengths of this study? What is alive and thrive strategies? Is it been supported by the government or NGO? Does the mothers for the child also working thus more fathers have to help their partners?

4) What is the anticipate plan after such findings?

Reviewer #2: Thank you for the opportunity to review this manuscript. This is an interesting topic for nursing research.

Background:

Please provide information to highlight the significance of the study. Explain the father’s role in child feeding and how the father can influence the nutritional outcomes among children. Provide the association or relationship between malnutrition with father involvement in child feeding with literature support.

Methods:

My main concern is the questionnaire used for the study. Detailed description is needed for the development and validation of the questionnaire. It is not sufficient with one sentence “Literature is reviewed to develop a questionnaire (23).”

The questionnaire is consisted of 4 parts: Father involvement, Knowledge, attitudes. Culture of Fathers about Father Involvement in Child Feeding.

Please provide information on the number of items, the scoring methods and interpretation of the scores for each part of the questionnaire.

Item 3 to 9 for knowledge questions are more related to practices towards father involvement in child feeding as we cannot expect a dichotomous response for these questions and it is not appropriate to assess knowledge, for instance: Item no. 3 – “fathers involvement in child feeding in providing advice or suggestions on diet” and item no.6 – “fathers involvement in child feeding in prepare and cooking a meal for a child.”

Apart from that, items for father involvement activities in child feeding (assist mother with household chores, accompanying mother for child health clinics, allowing other family members/relatives to support the mother after delivery, transporting the child to health clinics, looked after your child when the mothers were not around). These questions are more related to activities of looking after the child, shar burden of the wife, not measure father involvement in child feeding directly.

Is it possible multiple submissions as the father might have two children range 6 months to 24 months?

Please provide justification of pilot test with 5% of sample size.

Please provide explanation on types of bivariate analysis and multivariate logistic regression analysis.

Results:

Please provide justification for 100% response rate.

Suggest to remove figure 1, 2 and 3.

Discussion:

The authors have discussed the study results and have attempted to link it with previous studies. However, the result was not been analyzed and argued critically.

Please provide description on “Alive &Thrive's activities in Ethiopia” as background information.

Overall comments:

This paper requires proof-reading as there are grammatical and sentence structure errors. Citing and referencing skills can be further improved. There are some claims that are not having references. As a reader, I had difficulties to follow which of the sentences / ideas were from the references and which were authors own ideas. I hope the authors be more precise with the referencing.

Reviewer #3: General Comments

Language – grammatical and editorial issues all over the paper.

The manuscript needs careful revision by a proof-reader as there are numerous grammatical

errors, missing words, and errors in tense which make it sometimes difficult to read easily.

Background:

The background overall well written. Just wonder.,

Paragraph 4 - Can the stated information linked significantly with father involvement in child feeding?

Is the father involvement truly important in determining a child development especially in an underdeveloped country?

Ensuring a 'good food' for their family is a big task and timely responsible.

To me, a good socioeconomic status is an answer for all particularly in country like Ethiopia, Kenya etc.

Method and Design

Study Design and Settings

Paragraph 1 Line 3 - the information given a least related with the study and hard for reader to understand the geography of the study setting, It would be good if the author can provide i.e. the setting map.

Participants and Sampling

Paragraph 1 - is repetition (L1-4) and (L4-8)

A good and systematic random sampling method had been applied.

Measurement

A questionnaire developed - was validated.

Data Quality Assurance

Well mentioned

Suggestion: the writing is too long can be rewrite and merge the paragraph 1 and 3.

Data Management and Analysis

The descriptive result should be concise i.e., a repetitive information in text, figures and tables should be avoided.

Figure 1-3 can be deleted; the same information can be found in the bottom of Table 2-4 and in the text.

Results

Socio-Demographic characteristics

The text can be shortened, all information from Table 1 is self-explanatory.

Information of Respondents about Father Involvement in Child Feeding

Is this data only reported in text? The title Information of Respondents about Father Involvement in Child Feeding seem to be confused with Table 5 and its text.

Knowledge of Fathers about Father Involvement in Child Feeding

Is the item listed in the Table 2 portrayed knowledge of fathers about father involvement? i.e. fathers involvement in child feeding in social and emotional support; fathers involvement in child feeding in sharing workload and health etc. Wonder such items can its represented father’s knowledge on father involvement.

The Attitude of Father about Father Involvement in Child Feeding

Table 3, explained on the fathers’ attitude, Figure 2 is redundant.

Culture of Fathers about Father Involvement in Child Feeding

Table 4 can explain the Culture of Fathers.

The Magnitude of Father Involvement in Child Feeding

Please check

Factors Associated with Father Involvement in Child Feeding

Table’s title can be revise i.e. Table 6: Fathers involvement in child feeding and Its’ Association Factor. Please write only concise text.

Discussion

This part is monotonous and sounded in one unvarying tone, would be better to have a linkage between the sentences or points, discussion should be more in depth, not just reporting and compared deadly with other.

Conclusion

Well conclude by restating the purposes of the study with summarized results.

and conclusion that can early predict.

References

Please be consistent.

Check-up: Reference no 8, 9,17,19,21,22, 23, 24, 28

---

## [Author Response · Author response to Decision Letter 0]

31 Jul 2022

RESPONSE TO EDITOR AND REVIEWERS

RESPONSE TO EDITOR

RESPONSE: Thank you for coordinating the review process and fruitful comments. We have revised the manuscript and addressed Reviewer’s comments.

Journal Requirements: 

COMMENT: When submitting your revision, we need you to address these additional requirements. 1. Please ensure that your manuscript meets PLOS ONE's style requirements, including those for file naming.

RESPONSE: We have checked and attest that all formatting and style requirements have been met and revised based on the guideline 

COMMENT: 2. You indicated that you had ethical approval for your study. In your Methods section, please ensure you have also stated whether you obtained consent from parents or guardians of the minors included in the study or whether the research ethics committee or IRB specifically waived the need for their consent.

RESPONSE: we have assured for you that our all-study participant were adults

COMMENT: 3. Thank you for stating the following financial disclosure: 

 [No funders]. At this time, please address the following queries:

RESPONSE: We have added “The authors received no specific funding for this work.”

COMMENT: 4. Your ethics statement should only appear in the Methods section of your manuscript. If your ethics statement is written in any section besides the Methods, please move it to the Methods section and delete it from any other section. Please ensure that your ethics statement is included in your manuscript, as the ethics statement entered into the online submission form will not be published alongside your manuscript. 

RESPONSE: We have checked and ensured that ethical statement only appear in the methods section of our manuscript

COMMENT: 5. Please ensure that you refer to Figure 3 in your text as, if accepted, production will need this reference to link the reader to the figure.

RESPONSE: We have removed fig 3 because we accept suggestion from reviewer 2 to remove figure 1, 2 and 3.

COMMENT: 6. Please include your tables as part of your main manuscript and remove the individual files. Please note that supplementary tables (should remain/ be uploaded) as separate "supporting information" files.

RESPONSE: We have removed table as individual file and we include it in our main manuscript

Additional Editor Comments:

COMMENT: Please ensure the manuscript has undergo proof reading before resubmission 

RESPONSE: We have done it before resubmission.

RESPONSE TO REVIEWER 1:

RESPONSE: We thank the reviewer for kind words. We have revised our manuscript based on the comments as described below.

REVIEWER COMMENT: 1) Background - is there any studies as such has been identified in the world health report? The write up needs to be reword and revised such as too many dashes in page 4. It needs to be written clearly

RESPONSE: There is no studies has been identified in the world health report. We paraphrase it and we removed those inappropriate dash in page 4

REVIEWER COMMENT: 2) methods section, how did the author reach the community? Please explain. 

RESPONSE: We have reached to community as

Thee name and address of fathers having children aged 6 months to 2 years were specified and their location was identified in collaboration with the kebele’s health extension workers and health development army leaders, identified fathers were then interviewed in their home.

REVIEWER COMMENT: What is the response rate? 

RESPONSE: response rate refers to the number of people who completed your survey divided by the number of people who make up the total sample group.

REVIEWER COMMENT: What is kebeles in page 5.

RESPONSE: thank you for kind question. 

Ethiopia is administratively divided into four levels: regions, zones, woredas (districts) and kebele. Woredas are divided into kebele, municipalities. This is the smallest administrative division.

REVIEWER COMMENT: - With the questionnaire that is used has it been validated especially the KAP or etc. Please cite it. 

RESPONSE: the questionnaire was validated (27) and further more we have done pretest since pre-tests also provide the most direct evidence for the validity of the questionnaire; More over to determine the content validity of the questionnaire, the collected views, as well as comments experts in the fields of nutrition, biostatics, and epidemiology, were taken.

REVIEWER COMMENT: If pretest done, is there any variables been removed?

RESPONSE: We have done pre test and we revised the questionnaire based on the finding of pretest. We didn’t encounter removal of variable but we revised it for understandability, the wording, logic, and skip pattern etc, 

REVIEWER COMMENT: For data analysis, author highlighted double entry. Did you do that and what percentage is the discrepancy?

RESPONSE: We have done double data entry to ensure the integrity of the captured data. We ensured that Items in the double data entry are similar with the initial data entry.

REVIEWER COMMENT: - Ethical approval?

RESPONSE: We have revised Ethical approval in the methods section of our manuscript as

Ethical approval for this study was obtained from the Institution Review Board (IRB) of Zemen Post Graduate College. A permission letter from the Antsokia Gemza Woreda Health office and selected kebeles administrations were informed about the study. Data were collected after explaining the study, the benefit of the study, the research purpose orally, and getting verbal informed consent from each participant. 

REVIEWER COMMENT: 3) the discussion is too long.

RESPONSE: We have revised it based on your comment

REVIEWER COMMENT: What are the limitations and strengths of this study? 

RESPONSE: We have revised it as Limitation of the study

Since the study is cross-sectional it may not demonstrate direct cause and effect between dependent and independent variables. Another the limitation is possible to social desirability bias despite caution were used in order to reduce it. However, our study has some strength. As Ethiopian national nutrition program is targeting children and mothers to overcome malnutrition, this study tried to address father involvement in child feeding practices which if promoted could improve the overall health of children.

REVIEWER COMMENT: What is alive and thrive strategies? 

RESPONSE: We have been revised it and provide description on “Alive &Thrive's activities in Ethiopia” as background information. 

Alive &Thrive's activities in Ethiopia are designed to engage fathers in child feeding by identifying the strategies that seem to make these programs work to ensure that fathers can play an active role in improving child feeding practice by applying 6 strategies such as grabbing their attention with emotion, ease the way by busting stereotypes, find fathers where they already are, “provide crystal-clear direction” for actions fathers can take, give fathers practice and show fathers a benefit that they care about ( 22).

REVIEWER COMMENT: Is it been supported by the government or NGO?

RESPONSE: Alive & Thrive (A&T) funded by the Bill & Melinda Gates Foundation to improve infant and young child nutrition by increasing rates of exclusive breastfeeding and improving complementary feeding practices. 

REVIEWER COMMENT: Does the mothers for the child also working thus more fathers have to help their partners?

RESPONSE: We didn’t ask from mothers’ perspective; we would like to recommend for future researchers to gather information from both fathers and mothers’ perspective by including mothers as study subjects.

REVIEWER COMMENT: 4) What is the anticipate plan after such findings?

RESPONSE: We are glad to inform you that we have plan to present in conference at regional level and we invited to present it on July 26-28/2022. Here we attach their invitation letter.

 Congratulations!!!

Thank you, very much dear researchers, for submitting your abstract/s for consideration at APHI-BDU-CMHS joint international conference held from July 28-30/2014. All abstracts reviewed by several potential independent reviewers and status of your abstract/s is/are determined. Based on the reviews report, we are likely to accept your abstract/s for oral presentation in the conference, providing that you are kindly asked to prepare 15-20 slides following the template and present for 15 minutes.

Note: We sincerely request that all presenters collect and provide their travel receipts. Presenters who come by airline need only bring a passing board, a copy of their ticket, and bank receipts.

____//_____

*Tadesse Hailu Jember (PhD)

RESPONSE TO REVIEWER 2:

RESPONSE: We thank the reviewer for kind words. We have revised our manuscript based on the comments as described below.

REVIEWER COMMENT: Reviewer #2: Thank you for the opportunity to review this manuscript. This is an interesting topic for nursing research.

RESPONSE: We thank the reviewer for kind words. We have revised our manuscript based on the comments as described below.

Background:

REVIEWER COMMENT: Please provide information to highlight the significance of the study. 

RESPONSE: The finding of this study will be helpful for policymakers (whether governmental or Non-Governmental Organizations working on child feeding practices) to design evidence-based alternative strategies. It is also used for researchers as a baseline for future related studies. It is also a key for health professionals for providing evidence-based counseling on the study area. 

REVIEWER COMMENT: Explain the father’s role in child feeding and how the father can influence the nutritional outcomes among children. Provide the association or relationship between malnutrition with father involvement in child feeding with literature support.

RESPONSE: We have revised it as

Studies reveal a strong correlation between dads' substantial involvement in infant feeding activities and nutritional diversification (10). and families with involved fathers had significantly better breast-feeding practices than those with less involved fathers, such as by attending breast feeding sessions with moms, participating in decision-making, and supporting the mother around the house (11-12). Another community-based participatory intervention study revealed that involving both parents equally and actively in child-feeding is a promising strategy for preventing childhood obesity (13)

Methods:

REVIEWER COMMENT: My main concern is the questionnaire used for the study. Detailed description is needed for the development and validation of the questionnaire. It is not sufficient with one sentence “Literature is reviewed to develop a questionnaire (23).”

RESPONSE: We have revised it as

Literature is reviewed to develop a questionnaire and it was validated (27) and further more we have done pretest since pre-tests also provide the most direct evidence for the validity of the questionnaire. More over to determine the content validity of the questionnaire, the collected views, as well as comments experts in the fields of nutrition, biostatics, and epidemiology, were taken.

REVIEWER COMMENT: The questionnaire is consisted of 4 parts: Father involvement, Knowledge, attitudes. Culture of Fathers about Father Involvement in Child Feeding. Please provide information on the number of items, the scoring methods and interpretation of the scores for each part of the questionnaire.

RESPONSE: 

Fathers’ involvement in child feeding practice was assessed using 22 items with five major domain such as shared decision making in child feeding practice (6 items), providing physical support to the mother (5 items), providing psychosocial support (3 items), providing financial and resource support (5 items) and workload sharing (3 items). From these 22 questions directed to the fathers, variable scores of 12 and above were considered as good fathers’ involvement while variable scores of below 12 were considered as poor fathers’ involvement in child feeding practices (28).

Knowledge of fathers towards fathers’ involvement in child feeding was also assessed using 9 items that were directed to the fathers. The questions asked had the following components asked in such a way that do you know about; baby start complementary feeding at 6 months, recommended duration of breastfeeds 24 months or more, have role in providing advice on diet, have role in farming/gardening nutritious food, have role in financial support to buy nutritious food, have role in cooking a meal for a child, have role in accompaniment to medical appointments/growth monitoring, have role in in social and emotional support and have role to help child feeding mothers with house hold chores. Fathers who scored five and above among the listed nine items were considered as having good knowledge towards fathers’ involvement in child feeding practices while those who scored below five were considered as having poor knowledge towards fathers’ involvement in child feeding practices (29).

The responses of fathers about attitude towards father involvement in child feeding were in the form of 5-point Likert scale, wherein a score of +2 meant they strongly agreed to the statement, +1 meant agreed, 0 neutral feeling to the statement, -1 meant disagreed to the statement and -2 was the score for strongly disagree. The total score was calculated and was transformed into mean ‘percent score’ by dividing the score with possible maximum score and multiplied by 100. Scores for attitude was categorized as negative [0-<=60%] and positive [>60%] (29).

REVIEWER COMMENT: Item 3 to 9 for knowledge questions are more related to practices towards father involvement in child feeding as we cannot expect a dichotomous response for these questions and it is not appropriate to assess knowledge, for instance: Item no. 3 – “fathers’ involvement in child feeding in providing advice or suggestions on diet” and item no.6 – “fathers involvement in child feeding in prepare and cooking a meal for a child.”

RESPONSE: To the best of our knowledge, framing of knowledge items possible in two section such as multiple choice and yes or no items. it seems practice because we didn’t write whole part of question to reduce space, but it misleads readers. So that we have revised all knowledge questions as it was.

REVIEWER COMMENT: Apart from that, items for father involvement activities in child feeding (assist mother with household chores, accompanying mother for child health clinics, allowing other family members/relatives to support the mother after delivery, transporting the child to health clinics, looked after your child when the mothers were not around). These questions are more related to activities of looking after the child, shar burden of the wife, not measure father involvement in child feeding directly.

RESPONSE: We have been assured for you that father involvement is broad and we have considered father involvement in five broad domains such as 1. shared decision making in child feeding practice, 2 Providing Physical support to the mother, 3. Providing psycho-social support, 4. Providing Financial and resource support, 5. Workload sharing and promoting optimal child feeding practices. Those listed activities are directly or indirectly linked with father involvement in child feeding.

REVIEWER COMMENT: Is it possible multiple submissions as the father might have two children range 6 months to 24 months?

RESPONSE: There is no possibility for multiple submissions even if father have two children range 6 months to 24 months, we already asked him about index child.

REVIEWER COMMENT: Please provide justification of pilot test with 5% of sample size.

RESPONSE: We have done pretest on 5% of sample size because due to low resource and time.

REVIEWER COMMENT: Please provide explanation on types of bivariate analysis and multivariate logistic regression analysis.

RESPONSE: We have revised it as 

Independent variables with a P-value < of 0.25 in the bivariate analysis entered into the multivariable logistic regression in order to control the possible effect of confounders using the backward likelihood regression variable selection method.

Results:

REVIEWER COMMENT: Please provide justification for 100% response rate.

RESPONSE: This 100% response rate was achieved by multiple contacts; a person was deemed non-responsive if absent after three consecutive visits.

REVIEWER COMMENT: Suggest to remove figure 1, 2 and 3.

RESPONSE: We have removed figure 1, 2 and 3

Discussion:

REVIEWER COMMENT: The authors have discussed the study results and have attempted to link it with previous studies. However, the result was not been analyzed and argued critically.

RESPONSE: We have revised it based on your comment

REVIEWER COMMENT: Please provide description on “Alive &Thrive's activities in Ethiopia” as background information.

RESPONSE: We have been revised it and provide description on “Alive &Thrive's activities in Ethiopia” as background information.

Alive &Thrive's activities in Ethiopia are designed to engage fathers in child feeding by identifying the strategies that seem to make these programs work to ensure that fathers can play an active role in improving child feeding practice by applying 6 strategies such as grabbing their attention with emotion, ease the way by busting stereotypes, find fathers where they already are, “provide crystal-clear direction” for actions fathers can take, give fathers practice and show fathers a benefit that they care about ( 22).

Overall comments:

REVIEWER COMMENT: This paper requires proof-reading as there are grammatical and sentence structure errors. 

RESPONSE: We have revised our manuscript based on the comments for grammatical and sentence structure errors. 

REVIEWER COMMENT: Citing and referencing skills can be further improved. There are some claims that are not having references. As a reader, I had difficulties to follow which of the sentences / ideas were from the references and which were authors own ideas. I hope the authors be more precise with the referencing.

RESPONSE: We have revised our referencing to reduce any confusion.

RESPONSE TO REVIEWER 3:

RESPONSE: We thank the reviewer for kind words. We have revised our manuscript based on the comments as described below.

Reviewer #3: General Comments

REVIEWER COMMENT: Language – grammatical and editorial issues all over the paper.

The manuscript needs careful revision by a proof-reader as there are numerous grammatical errors, missing words, and errors in tense which make it sometimes difficult to read easily.

RESPONSE: We thank the reviewer for kind words. We have revised our manuscript based on the comments for grammatical and editorial issues.

Background:

REVIEWER COMMENT: The background overall well written. Just wonder.,

Paragraph 4 - Can the stated information linked significantly with father involvement in child feeding?

RESPONSE: We thank the reviewer for kind words. Stated information in paragraph 4 does not have linked with father involvement in child feeding. Even if it has no direct link with it, we just providing general information on malnutrition globally, sub-Saharan Africa and Ethiopia. If you suggest us to delete it, please let us know.

REVIEWER COMMENT: Is the father involvement truly important in determining a child development especially in an underdeveloped country? Ensuring a 'good food' for their family is a big task and timely responsible. To me, a good socioeconomic status is an answer for all particularly in country like Ethiopia, Kenya etc.

RESPONSE: we have considered father involvement in five broad domains such as 1. shared decision making in child feeding practice, 2 Providing Physical support to the mother, 3. Providing psycho-social support, 4. Providing Financial and resource support, 5. Workload sharing and promoting optimal child feeding practices 

so that as you stated above good socioeconomic status is an answer for all particularly in developing country, father involvement also includes the above broad domains which include Providing Financial and resource support which has indirectly linkage with determining a child development.

Method and Design

REVIEWER COMMENT: Study Design and Settings

Paragraph 1 Line 3 - the information given a least related with the study and hard for reader to understand the geography of the study setting, It would be good if the author can provide i.e. the setting map.

RESPONSE: We have revised it and provide map of study setting

REVIEWER COMMENT: Participants and Sampling

Paragraph 1 - is repetition (L1-4) and (L4-8)

RESPONSE: We have deleted (L1-4) to reduce repetition 

REVIEWER COMMENT: A good and systematic random sampling method had been applied.

RESPONSE: Thank you for kind words.

REVIEWER COMMENT: Measurement A questionnaire developed - was validated.

RESPONSE: Yes, developed questionnaire was validated.

Literature is reviewed to develop a questionnaire and it was validated (27) and further more we have done pretest since pre-tests also provide the most direct evidence for the validity of the questionnaire. More over to determine the content validity of the questionnaire, the collected views, as well as comments experts in the fields of nutrition, biostatics, and epidemiology, were taken.

REVIEWER COMMENT: Data Quality Assurance

Well mentioned Suggestion: the writing is too long can be rewrite and merge the paragraph 1 and 3.

RESPONSE: Thank you for kind words. We have revised and merged the paragraph 1 and 3 as.

A pre-test was done on 5% (20) of the sample size before the actual data collection. Amendments were made on questions after pre-testing. The questionnaire was prepared first in English and then translated to the local language of Amharic and back-translated to English by a third person who was native to Amharic and had experience in translation. The reliability tests were checked using Cronbach's alpha of 0.7 as the cut-off point to assess the internal consistency of the research instrument. The overall Cronbach's Alpha Coefficient value for the data collecting instrument was found to be 0.827 indicating the acceptability of the scale for further analysis. Hosmer and Lemeshow's test value was checked for its model fitness and its result is 0.833 indicating the model was fit.

REVIEWER COMMENT: Data Management and Analysis

The descriptive result should be concise i.e., a repetitive information in text, figures and tables should be avoided. Figure 1-3 can be deleted; the same information can be found in the bottom of Table 2-4 and in the text.

RESPONSE: We have revised it and we removed Figure 1-3 to reduce redundancy

Results

REVIEWER COMMENT: Socio-Demographic characteristics

The text can be shortened, all information from Table 1 is self-explanatory.

RESPONSE: We revised it as

 A total of 408 fathers who had children between 6 to 23 months were enrolled in the study with a response rate of100%. The mean age of the study participants was 36.8 with a standard deviation (SD) of 8.892 years and the majority of them, 334 (81.9%) lived in rural areas.

REVIEWER COMMENT: Information of Respondents about Father Involvement in Child Feeding Is this data only reported in text? The title Information of Respondents about Father Involvement in Child Feeding seem to be confused with Table 5 and its text.

RESPONSE: Yes, this part is only reported in text. We have revised it as Source of Information of Respondents about Father Involvement in Child Feeding. Table 5 has different idea and it is about items report to say good or poor father involvement.

REVIEWER COMMENT: Knowledge of Fathers about Father Involvement in Child Feeding Is the item listed in the Table 2 portrayed knowledge of fathers about father involvement? i.e., fathers’ involvement in child feeding in social and emotional support; fathers’ involvement in child feeding in sharing workload and health etc. Wonder such items can its represented father’s knowledge on father involvement.

RESPONSE: it seems practice because we didn’t write whole part of question to reduce space, but it misleads readers. So that we have revised all knowledge questions as it was. As we provide information about father involvement in five broad domains such as 1. shared decision making in child feeding practice, 2 Providing Physical support to the mother, 3. Providing psycho-social support, 4. Providing Financial and resource support, 5. Workload sharing and promoting optimal child feeding practices. So that we want to know fathers’ knowledge on whether they have role on those listed domains which are directly or indirectly linked with father involvement in child feeding.

REVIEWER COMMENT: The Attitude of Father about Father Involvement in Child Feeding Table 3, explained on the fathers’ attitude, Figure 2 is redundant.

Culture of Fathers about Father Involvement in Child Feeding Table 4 can explain the Culture of Fathers. The Magnitude of Father Involvement in Child Feeding Please check

RESPONSE: We have revised it and we removed Figure 1-3 to reduce redundancy

REVIEWER COMMENT: Factors Associated with Father Involvement in Child Feeding. Table’s title can be revise i.e., Table 6: Fathers involvement in child feeding and Its’ Association Factor. Please write only concise text.

RESPONSE: We have revised it as Table 6: Multivariable logistic regression model for fathers’ involvement in child feeding practice in Antsokia Gemza Wereda, North East Ethiopia, 2022.

Discussion

REVIEWER COMMENT: 

This part is monotonous and sounded in one unvarying tone, would be better to have a linkage between the sentences or points, discussion should be more in depth, not just reporting and compared deadly with other.

RESPONSE: We have revised it based on your comment

Conclusion

REVIEWER COMMENT: Well, conclude by restating the purposes of the study with summarized results and conclusion that can early predict.

RESPONSE: We have revised it as

The finding of this study showed that good father involvement in child feeding was low (43.1%), since the world’s Father’s Day 2019 report, which aims to facilitate an enabling environment where men take on 50% of involvement. Better (secondary and higher) educational status, having ever heard (information), urban residence, male sex of the youngest child, first birth order of the youngest child, good knowledge, positive attitude, and good culture were predictors of good fathers’ involvement in child feeding. The purpose of this study was to assess fathers’ involvement in child feeding practice and associated factors among fathers having children aged 6 to 24 months Hence, it is important to promote fathers’ involvement in child feeding practices through promoting their education, availing information sources, improving their knowledge, attitude and culture of fathers about child feeding practices. Future researchers should better conduct studies by using a mixed methodology and including females as potential participants to ensure fathers' involvement from women's perspective.

References

REVIEWER COMMENT: Please be consistent. Check-up: Reference no 8, 9,17,19,21,22, 23, 24, 28

RESPONSE: We have revised our reference based on your comment and for further checkup reference no 8: deleted based on other reviewer suggestion, 9: changed to 8, 17: changed to 20, 19: changed to 23, 21: changed to 25, 22: changed to 26, 23: changed to 27, 24: changed to 28, 28: changed to 31

---

## [Decision Letter · Decision Letter 1]

8 Sep 2022

PONE-D-22-14665R1Fathers Involvement in Child Feeding and Its Associated Factors Among Fathers Having Children Aged 6 to 24 Months in Antsokia Gemza Woreda, Ethiopia: Cross-Sectional StudyPLOS ONE

Dear Dr. Bogale,

Thank you for submitting your manuscript to PLOS ONE. After careful consideration, we feel that it has merit but does not fully meet PLOS ONE’s publication criteria as it currently stands. Therefore, we invite you to submit a revised version of the manuscript that addresses the points raised during the review process.

We look forward to receiving your revised manuscript.

Kind regards,

Khatijah Lim Abdullah, DClinP, MSc., BSc

Academic Editor

PLOS ONE

Journal Requirements:

Additional Editor Comments:

Dear Authors

Thank you for the revised manuscript. However, there are minor issues that need to be addressed as per reviewer comments and the need for proof reading

Reviewers' comments:

Reviewer's Responses to Questions

**Comments to the Author**

1. If the authors have adequately addressed your comments raised in a previous round of review and you feel that this manuscript is now acceptable for publication, you may indicate that here to bypass the “Comments to the Author” section, enter your conflict of interest statement in the “Confidential to Editor” section, and submit your "Accept" recommendation.

Reviewer #1: All comments have been addressed

Reviewer #2: (No Response)

2. Is the manuscript technically sound, and do the data support the conclusions?

Reviewer #1: Partly

Reviewer #2: Yes

3. Has the statistical analysis been performed appropriately and rigorously? 

Reviewer #1: Yes

Reviewer #2: Yes

4. Have the authors made all data underlying the findings in their manuscript fully available?

Reviewer #1: No

Reviewer #2: Yes

5. Is the manuscript presented in an intelligible fashion and written in standard English?

Reviewer #1: Yes

Reviewer #2: No

6. Review Comments to the Author

Reviewer #1: It is not clear in Table 6, the income is less than 500 and etc. What is the minimum wage for a family should be? 500 is equivalent to how many dollars (USD)?

I don't understand in financial disclosure no funders but this work is supported by Bill and Melinda gates as well ?

Besides this study findings will be presented at conference, how is the data will be conveyed to the general public in Ethiopia?

Reviewer #2: Thank you for the opportunity to review the revision of the manuscript. This manuscript is very much improved and it is good to see that the authors have addressed the issues based on reviewers’ feedback. However, the validation of the questionnaire issue remains.

Methods:

Please provide clearer illustration of the map (Figure 1).

The key area of concern relates to the validation of the questionnaire. There is insufficient evidence provided to support the claims that the instruments used are reliable and valid.

Detailed description is needed for the development and validation of the questionnaire. “Literature is reviewed to develop a questionnaire and it was validated…”

Please provide psychometric properties of the instrument.

Overall comments:

It is strongly suggested to send the paper for proof-reading as there are grammatical and sentence structure errors.

7. PLOS authors have the option to publish the peer review history of their article (what does this mean?). If published, this will include your full peer review and any attached files.

Reviewer #1: No

Reviewer #2: No

---

## [Author Response · Author response to Decision Letter 1]

12 Sep 2022

RESPONSE TO EDITOR AND REVIEWERS

RESPONSE TO EDITOR

RESPONSE: Thank you for coordinating the review process and fruitful comments. We have revised the manuscript and addressed Reviewer’s comments.

Journal Requirements: 

COMMENT: Please review your reference list to ensure that it is complete and correct. If you have cited papers that have been retracted, please include the rationale for doing so in the manuscript text, or remove these references and replace them with relevant current references. Any changes to the reference list should be mentioned in the rebuttal letter that accompanies your revised manuscript. If you need to cite a retracted article, indicate the article’s retracted status in the References list and also include a citation and full reference for the retraction notice.

RESPONSE: We have reviewed our reference list and we have got it complete and correct

Additional Editor Comments:

COMMENT: Dear Authors, Thank you for the revised manuscript. However, there are minor issues that need to be addressed as per reviewer comments and the need for proof reading

RESPONSE: We thank the reviewer for kind words and for fruitful comments. We have revised our paper by correcting grammatical and sentence structure errors.

RESPONSE TO REVIEWER 1:

Reviewer #1: All comments have been addressed

RESPONSE: We thank the reviewer for kind words. We have revised our manuscript based on the comments as described below. 

REVIEWER COMMENT: Reviewer #1: It is not clear in Table 6; the income is less than 500 and etc. What is the minimum wage for a family should be? 500 is equivalent to how many dollars (USD)?

RESPONSE: We thank the reviewer for kind words and comments. We would like to notify for you that we add the term Ethiopian birr to make clarity for the income. Now a days 52.811 Ethiopian birr is equivalent to 1 USD dollars. So that 500 Ethiopian birr is Equivalent to 9.467 USD dollars. In Ethiopia, the household is deemed as living in poverty if the per capita consumption is less than equal Birr 3781, so that minimum wage for a family to classify as poor is less than 500 birrs.

REVIEWER COMMENT: I don't understand in financial disclosure no funders but this work is supported by Bill and Melinda gates as well?

RESPONSE: We thank the reviewer for kind words and comments. We have not received specific funding starting from proposal development to report writing. We ensured for you that, we have not received specific fund from Bill and Melinda gates. 

REVIEWER COMMENT: Besides this study findings will be presented at conference, how is the data will be conveyed to the general public in Ethiopia?

RESPONSE: We thank the reviewer for kind words and comments. We will have plan to contact Ethiopian Ministry of Health to use our data for policy development and to design intervention based on our finding.

RESPONSE TO REVIEWER 2:

PONE-D-22-14665_R1: Fathers Involvement in Child Feeding and Its Associated Factors Among Fathers Having Children Aged 6 to 24 Months in Antsokia Gemza Woreda, Ethiopia: Cross- Sectional Study 1 Thank you for the opportunity to review the revision of the manuscript. This manuscript is very much improved and it is good to see that the authors have addressed the issues based on reviewers’ feedback. However, the validation of the questionnaire issue remains.

RESPONSE: We thank the reviewer for kind words. We have revised our manuscript based on the comments as described below.

REVIEWER COMMENT: Please provide clearer illustration of the map (Figure 1).

RESPONSE: We thank the reviewer for kind words. We have been readjusted it based on your comment

REVIEWER COMMENT: - 

Methods: The key area of concern relates to the validation of the questionnaire. There is insufficient evidence provided to support the claims that the instruments used are reliable and valid. Detailed description is needed for the development and validation of the questionnaire. “Literature is reviewed to develop a questionnaire and it was validated…” Please provide psychometric properties of the instrument.

RESPONSE: we adapt the questionnaire with necessary modification from the study conducted in North East Nigeria (27) which aimed to asses fathers’ knowledge, attitude and practices of fathers on their role to support recommended IYCF practices. In that study the questionnaire preliminary draft questionnaire developed by the consultant was reviewed by SC. It was then pre-tested in the field, revised and completed for the field survey, but psychometric analysis was not performed. Furthermore, as we informed you before, we have done pretest since pre-tests also provide the most direct evidence for the validity of the questionnaire; More over to determine the content validity of the questionnaire, the collected views, as well as comments experts in the fields of nutrition, biostatics, and epidemiology, were taken. We thank the reviewer for kind words and such an interesting comment and we will consider your comment as a title for future study which entitled with tool validation on fathers Involvement in Child Feeding and Its Associated Factors by using Confirmatory factor analysis. 

REVIEWER COMMENT: Overall comments: It is strongly suggested to send the paper for proof-reading as there are grammatical and sentence structure errors.

RESPONSE: Thank You for fruitful comments. We have revised our paper by correcting grammatical and sentence structure errors.

---

## [Decision Letter · Decision Letter 2]

27 Sep 2022

PONE-D-22-14665R2Fathers Involvement in Child Feeding and Its Associated Factors Among Fathers Having Children Aged 6 to 24 Months in Antsokia Gemza Woreda, Ethiopia: Cross-Sectional StudyPLOS ONE

Dear Dr. Bogale,

Thank you for submitting your manuscript to PLOS ONE. After careful consideration, we feel that it has merit but does not fully meet PLOS ONE’s publication criteria as it currently stands. Therefore, we invite you to submit a revised version of the manuscript that addresses the points raised during the review process.

We look forward to receiving your revised manuscript.

Kind regards,

Khatijah Lim Abdullah, DClinP, MSc., BSc

Academic Editor

PLOS ONE

Journal Requirements:

Additional Editor Comments:

Dear Authors

Many thanks for the time and effort in revising the manuscript

Overall most of the comments have been addressed

However we noted that although Table 6 have been added the term Ethiopian birr to make clarity for the income, it was noted that

1) Table 1 should also have the added term Ethiopian birr to made clear on the household monthly income.

2) To include an NB that a) 52.811 Ethiopian birr is equivalent to 1 USD dollars. b) minimum wage for a family to classify as poor is less than 500 birrs.

There is also a need to proof read the manuscript before acceptance.

Reviewers' comments:

Reviewer's Responses to Questions

**Comments to the Author**

1. If the authors have adequately addressed your comments raised in a previous round of review and you feel that this manuscript is now acceptable for publication, you may indicate that here to bypass the “Comments to the Author” section, enter your conflict of interest statement in the “Confidential to Editor” section, and submit your "Accept" recommendation.

Reviewer #2: All comments have been addressed

2. Is the manuscript technically sound, and do the data support the conclusions?

Reviewer #2: Yes

3. Has the statistical analysis been performed appropriately and rigorously? 

Reviewer #2: Yes

4. Have the authors made all data underlying the findings in their manuscript fully available?

Reviewer #2: Yes

5. Is the manuscript presented in an intelligible fashion and written in standard English?

Reviewer #2: Yes

6. Review Comments to the Author

Reviewer #2: Thank you for the opportunity to review the second revision of the manuscript.

All the comments have been addressed. The paper will make an important contribution to the literature.

7. PLOS authors have the option to publish the peer review history of their article (what does this mean?). If published, this will include your full peer review and any attached files.

Reviewer #2: No

---

## [Author Response · Author response to Decision Letter 2]

28 Sep 2022

RESPONSE TO EDITOR AND REVIEWERS

RESPONSE TO EDITOR

Thank you for submitting your manuscript to PLOS ONE. After careful consideration, we feel that it has merit but does not fully meet PLOS ONE’s publication criteria as it currently stands. Therefore, we invite you to submit a revised version of the manuscript that addresses the points raised during the review process.

RESPONSE: Thank you for coordinating the review process and fruitful comments. We have revised the manuscript and addressed both editor and Reviewer’s comments.

Journal Requirements: 

COMMENT: Please review your reference list to ensure that it is complete and correct. If you have cited papers that have been retracted, please include the rationale for doing so in the manuscript text, or remove these references and replace them with relevant current references. Any changes to the reference list should be mentioned in the rebuttal letter that accompanies your revised manuscript. If you need to cite a retracted article, indicate the article’s retracted status in the References list and also include a citation and full reference for the retraction notice.

RESPONSE: We have reviewed our reference list and we have got it complete and correct

Additional Editor Comments:

COMMENT: 

Many thanks for the time and effort in revising the manuscript. Overall, most of the comments have been addressed. However, we noted that although Table 6 have been added the term Ethiopian birr to make clarity for the income, it was noted that

1) Table 1 should also have the added term Ethiopian birr to made clear on the household monthly income.

2) To include an NB that a) 52.811 Ethiopian birr is equivalent to 1 USD dollars. b) minimum wage for a family to classify as poor is less than 500 birrs.

RESPONSE: We thank the editor for kind words and for fruitful comments. We have revised it 

1) by adding word In Ethiopian birr in table 1 in household monthly income part.

2) by adding remember note below table 1 and table 6 as: a) 52.811 Ethiopian birr is equivalent to 1 USD dollars. b) minimum wage for a family to classify as poor is less than 500 birrs.

COMMENT: There is also a need to proof read the manuscript before acceptance.

RESPONSE: We thank the reviewer for kind words and for fruitful comments. We have revised our paper by correcting grammatical and sentence structure errors.

RESPONSE TO REVIEWER 1:

Reviewer's Responses to Questions

Comments to the Author

1. If the authors have adequately addressed your comments raised in a previous round of review and you feel that this manuscript is now acceptable for publication, you may indicate that here to bypass the “Comments to the Author” section, enter your conflict-of-interest statement in the “Confidential to Editor” section, and submit your "Accept" recommendation.

Reviewer #2: All comments have been addressed

RESPONSE: We thank the reviewer for kind words. 

2. Is the manuscript technically sound, and do the data support the conclusions?

Reviewer #2: Yes

RESPONSE: We thank the reviewer for kind words. 

3. Has the statistical analysis been performed appropriately and rigorously?

Reviewer #2: Yes

RESPONSE: We thank the reviewer for kind words.

 4. Have the authors made all data underlying the findings in their manuscript fully available?

Reviewer #2: Yes

RESPONSE: We thank the reviewer for kind words.

5. Is the manuscript presented in an intelligible fashion and written in standard English?

Reviewer #2: Yes

RESPONSE: We thank the reviewer for kind words.

6. Review Comments to the Author

Reviewer #2: Thank you for the opportunity to review the second revision of the manuscript.

All the comments have been addressed. The paper will make an important contribution to the literature.

RESPONSE: We thank the reviewer for kind words.

7. PLOS authors have the option to publish the peer review history of their article (what does this mean?). If published, this will include your full peer review and any attached files.

Do you want your identity to be public for this peer review? For information about this choice, including consent withdrawal, please see our Privacy Policy.

Reviewer #2: No

RESPONSE: We thank the reviewer for kind words.

---

## [Editor Report · Decision Letter 3]

10 Oct 2022

Fathers Involvement in Child Feeding and Its Associated Factors Among Fathers Having Children Aged 6 to 24 Months in Antsokia Gemza Woreda, Ethiopia: Cross-Sectional Study

PONE-D-22-14665R3

Dear Dr. Bogale,

We’re pleased to inform you that your manuscript has been judged scientifically suitable for publication and will be formally accepted for publication once it meets all outstanding technical requirements.

Kind regards,

Khatijah Lim Abdullah, DClinP, MSc., BSc

Academic Editor

PLOS ONE
---

## [Editor Report · Acceptance letter]

14 Oct 2022

PONE-D-22-14665R3 

Fathers Involvement in Child Feeding and Its Associated Factors Among Fathers Having Children Aged 6 to 24 Months in Antsokia Gemza Woreda, Ethiopia: Cross-Sectional Study 

Dear Dr. Bogale:

I'm pleased to inform you that your manuscript has been deemed suitable for publication in PLOS ONE. Congratulations! Your manuscript is now with our production department. 

Kind regards, 

on behalf of

Dr. Khatijah Lim Abdullah 

Academic Editor

PLOS ONE